# Effects of Different Salt Ion Concentrations in Immersion Vacuum Cooling on the Qualities of Spiced Chicken Drumsticks

**DOI:** 10.3390/foods11244063

**Published:** 2022-12-15

**Authors:** Di Zhou, Rui Song, Guofu Yi, Qingli Han, Huazhen Cai, Yawei Zhang, Yuxia Zhu

**Affiliations:** 1Anhui Heat-Sensitive Materials Processing Engineering Technology Research Center, School of Biological Science and Food Engineering, Chuzhou University, Chuzhou 239000, China; 2National Center of Meat Quality and Safety Control, College of Food Science and Technology, Nanjing Agricultural University, Nanjing 210095, China

**Keywords:** immersion vacuum cooling, cooling rate, low-temperature processing, salt ion concentration, drumsticks, nuclear magnetic resonance

## Abstract

The traditional immersion vacuum cooling of meats can result in product defects. To optimize these processes, different salt ion concentrations in the immersion solution (0%, 3%, 5%, and 7%) were assessed, in relation to the cooling rate, cooling loss rate, color, texture, moisture status, and microstructure of chicken drumsticks. The cooling rate at 5% salt ion concentration was the fastest and most similar to the conventional vacuum cooling method, which can reduce the central temperature of drumsticks from 75 to 25 °C in 15 min. Immersion vacuum cooling did not cause weight loss and the 5% salt ion concentration was the best for weight maintenance, which can increase the weight of drumsticks by 2.3%. The *L** and *b** values first decreased and then increased with increasing salt ion concentrations, but not significantly. Hardness gradually decreased, whereas the low-field nuclear magnetic data showed that the transverse relaxation time of free water (*T*_22_) in the drumsticks increased from 200.01 ms to 237.79 ms with increasing salt ion concentrations. Scanning electron microscopy images revealed irregular and smaller pores between the muscle fibers with increasing salt ion concentrations. The 5% salt ion concentration in the immersion solution during vacuum cooling was optimal as it increased the cooling rate and improved the edible quality without cooling loss. Thus, adjusting the salt ion concentration of the immersion solution is a feasible way to improve economic benefits and quality characteristics of meat products.

## 1. Introduction

Ready-to-eat meat products are prepared by adding a certain proportion of raw meat to water containing the prepared seasonings and spices, followed by heating and boiling [1,2]. These products retain the maximum nutrition of the original meat and possess a desirable flavor, color, and texture [3,4]. To improve their quality and microbial safety, these products should be cooled to a safe temperature as soon as possible after cooking. [5,6]. Consequently, many countries have devised restrictive regulations and recommendations for the cooling time of cooked meat products [7,8]. Several traditional cooling techniques, such as blast cooling, cold chamber cooling, and hydro-cooling, cannot meet the demands of joint meats owing to their low cooling rates, which increase the risk of product decay [2,9]. In recent years, there has been active research in the meat industry to develop new and effective cooling technologies. Among these, vacuum cooling (VC) has been found to provide superior cooling rates and homogeneity in meat products [3,10,11,12]. However, evaporation of water under low pressure from cooked food leads to an undesired loss of mass, causing poor taste for consumers, and less profits for manufacturers, which has limited the widespread application of the VC technology in the meat industry.

Immersion vacuum cooling (IVC) is an innovative modified VC method, which can cool meat products quickly when compared to traditional cooling methods, such as blast cooling and cold-water cooling. Compared with VC, IVC can effectively reduce the weight loss of meat caused by cooling and can improve the quality of cooling meat. IVC has been tested with various meat products, such as beef [13], sausages [14], and pork hams [7] and the results have been promising. However, as this method requires longer cooling times when compared to VC and is associated with quality changes in the meat, further research is required before it can be used widely. To address this issue, several modified methods have been explored to improve IVC, such as IVC combined with other cooling methods, immersion vacuum cooling with agitation, pressure control, and immersion solutions with different initial temperatures [15,16,17,18]. Guo et al. illustrated that bubbling IVC could accelerate the cooling of cooked meat, reduce weight loss and had no significant difference in moisture content, color and TPA compared with conventional IVC [19]. Liao et al. found that IVC with ultrasonic assistance (IVCUA) clearly reduced the cooling time compared to IVC, and there were no significant differences in mass loss, color and texture profile among all samples. However, IVCUA had a higher transverse relation time of bulk water (*T*_24_) and MRI proton densities [20]. The main optimization directions of the IVC improvement method include several aspects, namely reducing the loss of moisture from meat and accelerating the cooling rate, improving the microbial safety of meat, and ensuring the eating quality of meat [9]. Although some studies have been carried out on the improvement of IVC, there are still some problems, such as the cooling rate of IVC being not fast enough, the quality indicators such as color and texture of meat after cooling changing considerably, and there is a risk of introducing exogenous microorganisms. Therefore, the focus of future research should be to explore ways to reduce the weight loss rate, and ensure the edible quality of cooked meat products and fast cooling rate.

The immersion solution plays an important role in VC; its composition, temperature and changes under vacuum cooling significantly affect the cooling rate and meat quality. The effects of using different salt ion concentrations on the quality of stewed chicken drumsticks has not yet been assessed. No research has been conducted to assess the effects of immersion solution with different salt concentrations on the vacuum cooling of meat. The primary objective of this study was to investigate the effects of different salt ion concentrations in the immersion solution on the cooling rate, cooling loss rate, chromatism, and textural properties of stewed chicken drumsticks during IVC. The transverse relaxation times of low-field nuclear magnetic resonance (LF-NMR) and scanning electron microscopy (SEM) images were analyzed to determine the possible mechanisms of water exchange and structural changes. The results will provide valuable reference data to help further improve IVC methods for meat products.

## 2. Materials and Methods

### 2.1. Materials and Reagents

The chicken drumstick samples and spices including dried salt, cinnamon, prickly ash, anise, geranium, cumin, and ginger were purchased from a local supermarket in Chuzhou, China. Each drumstick weighed approximately 260 ± 10 g (common meat weight for specific selections) and all the samples were vacuum-packed and stored individually at −18 °C. The spices were stored at room temperature (approximately 25 °C) in vacuum packaging (nylon/polyethylene). Absolute ethyl alcohol, ethyl alcohol, glutaraldehyde solution, and tert-butanol were sourced from Nanjing Chemical Reagents Co. Ltd, Nanjing, China. All chemicals and reagents used were of analytical grade.

### 2.2. Preparation of the Immersion Solution

The spice formula was as follows: cinnamon 0.2%, prickly ash 0.2%, anise 0.15%, geranium 0.14%, cumin 0.16%, and ginger 0.16%, in relation to the water (*w*/*w*). Spices were placed in a filter bag (gauze, 15 cm × 20 cm) and boiled in 10 kg of water using an induction cooker set at 100 °C. The temperature was maintained at 100 °C (high temperature) for 10 min after boiling and then reduced to 60 °C (low temperature) after which the boiling was continued for 50 min. The filter bags were then removed, different weights of salt were added to the boiled water (3%, 5%, and 7% of the water weight), and immersion solutions of different salt ion concentrations were obtained. Finally, for the immersion solution, the solutions with different salt ion concentrations (0%, 3%, 5%, and 7%) were agitated evenly and cooled to 15 °C.

### 2.3. Sample Preparation

The spiced liquid was prepared first, according to the immersion solution preparation method described in Section 2.2; the salt concentration in the brine was 3%. The drumstick samples were thawed for 24 h at 4 °C and then washed and trimmed. Subsequently, the preprocessed drumsticks were cooked individually in the spiced liquid at a high temperature for 5 min and then at a low temperature for 15 min.

### 2.4. Vacuum Cooling

As a control treatment, the cooked drumstick samples were immediately placed in a stainless-steel container (16 × 20 cm) in a vacuum cooler (DCS-60, DaChang Cooler Equipment Engineering Co. LTD, Ganzhou, China) and the temperature was reduced from 75 to 25 °C. The vacuum cooler was equipped with a thermocouple, which was inserted into the meat without touching the bones to measure the core temperature of each drumstick. After the instrument door was closed, the pressure of the chamber was reduced to 4000 Pa within a few minutes and the final core temperature was set to 25 °C. Each sample consisted of five drumsticks, and there were three replicates of each sample. The weights of the drumstick samples before and after cooling were also recorded.

### 2.5. Immersion Vacuum Cooling

The cooked drumstick samples were immediately placed in a stainless steel container (16 × 20 cm) with one of the immersion solutions (0%, 3%, 5%, and 7% salt concentrations, as described in Section 2.2) deep enough to just cover each sample at a temperature of 15 °C (Figure 1). The container was then placed in a vacuum cooler (DCS-60, DaChang Cooler Equipment Engineering Co., LTD, Ganzhou, China) and the temperature was reduced from 75 to 25 °C. The vacuum cooler was equipped with a thermocouple which was used as described in Section 2.4. After the instrument door was closed, the pressure of the chamber was reduced to 4000 Pa within a few minutes and the final core temperature was set to 25 °C. Each sample consisted of five drumsticks, and there were three replicates of each sample. The weights of the drumstick samples before and after cooling were also recorded.

### 2.6. Cooling Rate

Changes to the drumstick core temperatures were recorded at intervals of 30 s during cooling, and temperature vs. time curves were plotted using the recorded data. Each cooling rate test was completed three times and one of the three parallel samples used as a representative.

### 2.7. Cooling Loss Rate

The cooling loss was calculated using the following formula:(1)$$R=\frac{m_{1}-m_{2}}{m_{1}}\times 100$$
where *R* is the cooling loss rate, *m*_1_ is the weight before cooling (g), and *m*_2_ is the weight after cooling (g).

### 2.8. Color Measurement

The color of the inside portion of the cooled drumsticks was obtained using the CIE *L*a*b** (*L**: lightness, *a**: red/green, and *b**: yellow/blue) system with a colorimeter (CR-400, Konica Minolta Ltd., Tokyo, Japan). Using the color of drumsticks before cooling as control, the color difference value Δ*E* was determined and calculated according to Qu’s method [21].

### 2.9. Textural Properties

The drumstick samples without skin were cut into 10 × 10 × 10 mm pieces and their hardness, chewiness, and elasticity were measured using a textural instrument (TA. XT2i, Stable Micro Systems Ltd., Godalming, UK). The operating parameters are listed in Table 1.

### 2.10. Low-Field Nuclear Magnetic Resonance

Low-field nuclear magnetic resonance (LF-NMR) determination was performed using the method of Zhou et al. [22] with slight modifications. Approximately 7 g of each sample was cut along the muscle fiber direction from the inner part of the drumstick (1 × 1 × 2 cm^3^) and wrapped with raw tape and placed in a glass tube (15 mm diameter). The tube was then inserted into the probe of a low-field NMR analyzer (MeSONMRI-3060VI. Suzhou, Niumag Electric Co., LTD, Suzhou, China), and the resonant frequency was set to 22.4 MHz, with a working temperature of 32 °C. The transverse relaxation time (*T*_2_) was measured using a Carr-Purcell-Meiboom-Gill (CPMG) pulse sequence with 8 scans, 3000 echoes, 5 s repetition time, 4000 ms waiting time and 200 μs gap between 90 and 180° pulse. The MultiExp Inv. Analysis software (Niumag Electric Co., LTD, Suzhou, China) was used to invert the attenuation curve. The water distribution and composition of each drumstick was assessed using the transverse relaxation times of *T*_2*b*_, *T*_21_, and *T*_22_ and their corresponding water populations, *P*_2*b*_, *P*_22_, and *P*_23_ (*T*_2*b*_ and *P*_2*b*_ correspond to bound water; *T*_21_ and *P*_21_ correspond to immobilized water; and *T*_22_ and *P*_22_ correspond to free water).

### 2.11. Scanning Electron Microscopy

Scanning electron microscopy (SEM) was used for microstructure analysis, and the samples were prepared according to the method of Han et al. [23] with slight modifications. Outer slices of the spiced drumstick (without skin) were subjected to SEM analysis. The samples (1 × 1 × 1 mm^3^) were fixed for 2 h in 25% glutaraldehyde. After washing three times with PBS (pH = 7.3), the samples were dehydrated using increasing concentrations of ethyl alcohol (50%, 70%, 80%, and 90%) for 15 min each, followed by the addition of tert-butanol and absolute ethyl alcohol to continue the dehydration for 30 min. This was repeated three times. The pretreated samples were then subjected to vacuum freeze-drying. Finally, SEM micrographs were obtained using a scanning electron microscope at 20 kV after being sprayed with a gold sputter coating(JSM-6510LV, Jeol, Tokyo, Japan). One of the three parallel samples was used as a representative for the analysis.

### 2.12. Statistical Analysis

Experiments were performed in triplicate (except for specially declared) for each sample. SPSS 17.0 (SPSS Inc., Chicago, IL, USA) was used to evaluate the effects of the different salt ion concentrations used for the IVC of the spiced drumstick samples on the cooling time, cooling loss rate, chromatism, textural properties, and LF-NMR in the one-way analysis of variance (ANOVA) with Tukey’s multiple comparison test (*p* < 0.05). Origin 8.5 (OriginLab Corp., Northampton, MA, USA) was utilized to draw all diagrams.

## 3. Results and Discussion

### 3.1. Cooling Rate

The core temperature curves for the drumsticks treated with different salt ion concentrations indicate that there was a distinct difference in the cooling rate between the VC and IVC methods (Figure 2). The cooling time was approximately 15 min with the VC, during which the core temperature of the drumstick dropped from 75 °C to 25 °C, which was the same as the IVC cooling time when the salt ion concentration of the soaking solution was 5%. However, the cooling rate of VC was faster than those of the IVC samples with 0%, 3%, and 7% ion concentrations. Different heat-transfer mechanisms led to the differences between the methods. The heat of the drumstick was quickly removed by water evaporation during VC. In contrast, thermal conduction of the interior and convection of the surface from the meat samples was found to be key during IVC [13]. The different cooling mechanisms ultimately led to differences in the cooling rates between VC and IVC.

The different salt ion concentrations in the immersion solutions had a significant influence on the cooling rates of the drumsticks during IVC (Figure 2). The fastest cooling rate occurred at a salt ion concentration of 5% in just 15 min, which is similar to the duration required with VC, whereas at salt ion concentrations of 0%, 3%, and 7%, the cooling was slower, being 19, 18, and 19 min, respectively. The heat-transfer mechanism of IVC involved water evaporation, thermal convection, and thermal conduction. During the entire IVC cooling process, water boils violently and evaporates at the beginning, and then the cooling mainly relies on thermal convection and thermal conduction [24]. The results showed that when the salt ion concentration increased from 0% to 5%, the core temperature of the drumsticks decreased rapidly. At the violent stage, the water evaporation occurs both in drumsticks and the surrounding water. The increase in salt concentration increased the thermal conductivity of brine in the violent stage [25], and therefore, the water in drumsticks evaporated quickly and cooled toa greater extent. In addition, in the later stage of thermal convection and thermal conduction, the increase in salt concentration reduced the specific heat capacity of brine, which also made cooling easier. However, as the salt ion concentration continued to increase to 7%, the cooling rate of the drumsticks began to slow down. Due to the increasing concentration of salt ions, the boiling point of brine also increased, making the evaporation of water more difficult, and therefore, the cooling rate decreased.

### 3.2. Cooling Loss Rate

The effects of the different salt ion concentrations in the immersion solutions on the cooling loss rates of the drumsticks are summarized in Figure 3. There was a significant difference in the cooling loss rates of the drumstick samples treated with VC and IVC, as they reached 7.8% with the VC, but the weight was augmented to varying degrees by the IVC; specifically, the 0%, 3%, 5%, and 7% samples gained 1.6%, 1.2%, 2.3%, and 1.8% in weight after IVC treatment, respectively. These results are similar to those of previous studies [10,26]. With the evaporation of free water, undesired weight loss was observed during VC. However, the IVC could offset this loss as the immersion solution could penetrate the drumsticks at the end of the cooling period. Therefore, the weight of the samples was increased with IVC but decreased with VC.

The different salt ion concentrations in the immersion solution affected the cooling loss of the drumsticks during IVC (*p* < 0.05) (Figure 3). The weight of the drumsticks first increased and then decreased with the increasing salt ion concentration. The maximum and minimum weight increase occurred at salt ion concentrations of 5% and 3%, respectively. Houska [27] and Schmidt [2] also found that with different treatments, the weight of meat products increased to different degrees after IVC. In IVC, the increase of moisture in meat was mainly due to the rise in vacuum chamber pressure after the end of cooling, and the pressure difference caused a proportion of brine to penetrate the sample, resulting in the enhancement of sample quality. Together with the SEM results (Figure 4), it was found that with the increase in salt concentration, the number of slits and pores between muscle fiber tissues in meat increased. Therefore, it is speculated that when the salt concentration of brine is lower than 5%, the salt concentration in brine and meat reaches the equilibrium state, the osmotic pressure is small, and the pores between muscle fibers are fewer. Therefore, only the final pressure difference can make a proportion of the immersion solution penetrate the meat. When the brine salt concentration was 5%, a greater number of large cavities in the drumstick tissue were suitable for the immersion solution to permeate. A certain concentration of salt ions in the muscle tissue can also appropriately improve the water retention of meat, and the combined effect makes the mass increase the most. When the brine salt concentration was 7%, the weight gain of drumsticks decreased. This may be due to the fact that although there are more pores between the muscle fibers, the pore structure also changes. Liao reported that different salt concentrations not only change the porosity of meat tissue structure, but also make the pore structure of some muscle tissues more complex and the tortuous rate of the channel higher [28]. Therefore, the more complex pore structure blocks the infiltration of immersion solution to some extent and limits the weight gain of drumsticks.

### 3.3. Color

The effects of the different salt ion concentrations in the immersion solutions on the colors of the drumsticks are presented in Table 2. The *L** values of the samples cooled with VC were significantly lower than those treated with IVC, whereas the *b** values were significantly higher (*p* < 0.05), which may be owing to water loss and the pigment accumulation of the drumsticks during VC, which is consistent with the results of Song et al. [11]. With IVC, the immersion solution permeated the drumsticks and made up for the water loss, which contributed to a higher *L** value. However, the pigments in the drumsticks were also diluted, leading to lower *b** values. 

The different salt ion concentrations in the immersion solutions used for IVC had no statistically significant effect on color (*p >* 0.05). However, the changes in *L** tended to first decrease and then increase. This may be owing to the moisture content of the samples and the movement of the water molecules between the drumsticks and the immersion solution.

As shown in Figure 3 and Table 2, when the salt ion concentration was 5%, the weight of drumsticks increased the most, whereas the value of *b** was the least. However, the results were reversed when the salt ion concentration was 3%. These results indicated that the trends for *b** showed a negative correlation with the cooling loss of the drumsticks. Similar results were reported in previous studies, which revealed that the *L** and *a** values of beef have a certain relationship with the moisture content of meat [29]. The lower the moisture content, the greater the accumulation of pigment on the surface of meat, and the lower are the *L** and *a** values. Δ*E* is related to the overall change in color. When Δ*E* is greater than 4.0, an obvious change in the color of samples can be observed [21]. Table 2 showed that the Δ*E* of drumsticks after VC was 6.32, indicating that the color difference can be significantly observed before and after cooling. The Δ*E* values of drumsticks in IVC groups were all less than 4.0, and the color difference was the most obvious in the 3% IVC sample. This may be due to the longer cooling time of the 3% IVC sample, more pigmentation, and the smallest increase in weight, so the drumsticks color difference was the most obvious. However, it is worth mentioning that the color changes of all IVC groups are within the acceptable range of consumers.

### 3.4. Textural Properties

Textural properties are important as they reflect meat product quality. The effects of the salt ion concentrations in the immersion solutions on the hardness, elasticity, and chewiness of the drumsticks are shown in Table 3. The hardness, elasticity, and chewiness were significantly lower with the IVC treatment when compared with VC (*p* < 0.05). This was because the surrounding solution infiltrated the drumsticks, and the results are consistent with those of Song et al. [11].

There were no statistically significant differences in the elasticity or chewiness of the drumsticks with the different salt ion concentrations in the immersion solution cooled by IVC (*p* > 0.5). However, the hardness value significantly declined (*p* < 0.5), which may be caused by changes in the moisture distribution, composition, and microstructures (Figure 4 and Figure 5) within the drumsticks. As shown in Figure 4 and Table 4, with an increase in the salt ion concentration, the *T*_22_ of drumsticks was also increased, which had a positive correlation with the changes in the hardness. Especially in the immersion solutions with 5% and 7% salt concentrations, more free water in drumsticks contributed to lower hardness, whereas less free and bound water led to higher hardness in immersion solutions with 3% salt concentrations. The results could also be explained based on the microscopic structure of the drumsticks (Figure 5). When the salt ion concentration was greater than 5%, more small pores appeared in the fibers and tissues of the drumsticks, which indicate that more water could permeate the drumsticks, causing lower hardness. Thus, in the process of IVC, the increase in free water is the primary cause for the changes in hardness [19,20].

### 3.5. Low-Field Nuclear Magnetic Resonance

Relaxometry has been extensively applied to analyze the water composition and mobility of meat and meat products and is highly correlated with chromatism, texture, and water holding capacity [30]. Low sample water fluidity values result in short NMR transverse relaxation times (*T*_2_); in contrast, they are longer with higher liquidity [31]. The distributions of *T*_2_ and their populations (*P*_2_) from drumsticks processed using different cooling methods are shown in Figure 4 and Table 4. Three relaxation components were obtained from the attenuation curves (Figure 4). The *T*_2*b*_ component (1.0–10.0 ms) was regarded as bound water, which is strongly related to macromolecules. The *T*_21_ component (11–100 ms) was identified as immobilized water, which was limited to the protein structure. The *T*_22_ component (110–400 ms) was free water, which emerged outside the myofibrillar protein [32]. The longer the relaxation time, the lower the binding degree of water and substrate, and the higher the degree of freedom. The smaller the peak area, the less relative water content there is in this part.

**Figure 4 foods-11-04063-f004:**
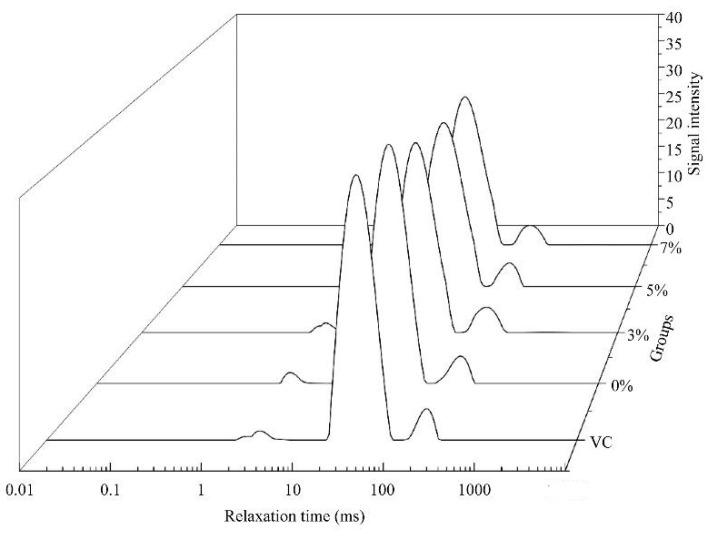
Three-dimensional waterfall plot showing the *T*_2_ changes in the chicken drumsticks in different cooling treatments generated with Origin 8.5. (*T*_2_: transverse relaxation time; VC: vacuum cooling; 0%, 3%, 5%, and 7%: salt ion concentrations in immersion solution during immersion vacuum cooling).

**Table 4 foods-11-04063-t004:** Effects of immersion solutions with different salt ion concentrations on the transverse relaxation time (*T_2_*) of drumsticks.

Salt Ion Concentrations ^1^	VC	0%	3%	5%	7%
*T*_2*b*_ (ms)	2.67 ± 0.05 ^a^	2.10 ± 0.13 ^a^	1.91 ± 0.80 ^a^	2.15 ± 0.72 ^a^	2.78 ± 0.12 ^a^
*T*_21_ (ms)	28.34 ± 0.71 ^b^	31.44 ± 1.14 ^a^	30.14 ± 1.21 ^ab^	32.25 ± 0.55 ^a^	32.11 ± 1.64 ^a^
*T*_22_ (ms)	200.01 ± 1.97 ^d^	219.64 ± 7.16 ^c^	228.69 ± 7.30 ^b^	235.97 ± 5.54 ^a^	237.79 ± 1.97 ^a^
*P*_2*b*_ (%)	2.71 ± 0.70 ^a^	2.50 ± 0.10 ^a^	2.66 ± 0.16 ^a^	2.35 ± 0.27 ^a^	2.25 ± 0.21 ^a^
*P*_21_ (%)	93.27 ± 1.10 ^a^	91.99 ± 1.34 ^b^	91.62 ± 3.04 ^b^	89.21 ± 1.80 ^c^	91.96 ± 2.21 ^b^
*P*_22_ (%)	4.02 ± 0.09 ^c^	5.51 ± 0.22 ^ab^	5.72 ± 0.18 ^ab^	8.44 ± 0.30 ^a^	5.79 ± 0.11 ^b^

^1^ Salt ion concentrations in the immersion solution during immersion vacuum cooling; VC: vacuum cooling; *T*_2_: transverse relaxation time; *P*_2_: *T*_2_ population. Data are expressed as the mean ± standard deviation of three replicates (*n* = 3). Different lowercase superscript letters in the same column indicate statistically significant differences (*p* < 0.05).

There was no significant difference between the transverse relaxation time of *T_2b_* and the peak ratio of *P*_2*b*_ in the drumsticks cooled by VC and IVC (*p* > 0.05), indicating that neither IVC nor VC could cause the change of the bound water in meat, which was consistent with the results of Jin [33]. The transverse relaxation time of *T*_21_, *T*_22_ and the peak ratio of *P*_22_ of IVC were higher than VC and the peak ratio of *P*_21_ was lower than VC, indicating that the proportion of free water in meat after IVC and the freedom of free water and immobilized water were higher than that of VC. This may be due to a large amount of free water and a small amount of immobilized water that migrated outward and was lost under the action of capillary force and water potential difference during VC. However, free water from the immersion solution could penetrate the drumsticks at the end of the cooling period.

As salt concentration increased from 0% to 5%, the transverse relaxation time of *T*_22_ in meat increased significantly, indicating that the freedom of free water in meat gradually increased. With the increase in salt ion concentration, the meat cooled at a faster rate during the initial cooling period, which means more water evaporated. During the water evaporation stage, water migrated in gaseous form, which also encourages the liquid water in the meat to move [28], resulting in an increase in freedom, which may explain the change of *T*_22_. As salt concentration increased from 5% to 7%, the transverse relaxation time of *T*_22_ also increased but not significantly. At the initial stage of cooling, the water evaporation of the 7% drumsticks was weakened, but in the subsequent cooling process, the osmotic pressure of salt concentration would promote the outward migration of free water in the meat, and these combined effects resulted in the increase of *T*_22_. It is noteworthy that the transverse relaxation time of *T*_22_ has a good positive correlation with the hardness of drumsticks, which is similar to the findings of Liao [28].

As salt concentration increased from 0% to 5%, the peak ratio of *P*_22_ in meat increased and the peak ratio of *P*_21_ decreased. IVC has been reported to enable the immersion of water in the late cooling and meat water exchange [19]. Therefore, we speculated that the effect of salt concentration on water proportion might be due to the combined effect of osmotic pressure of salt solution in the cooling process and pressure difference at the later stage of cooling, such that more bound water in meat was converted to free water. As the salt concentration increased from 5% to 7%, the peak ratio of *P*_22_ in meat decreased and the peak ratio of *P*_21_ increased. At this time, although osmotic pressure and pressure difference play a role, as shown in the SEM image, the pore complexity between muscle fibers increases, which limits the mutual conversion between bound and free water. In addition, we also found a correlation between the change rule of *P*_22_ and the cooling rate—the greater the peak ratio of *P*_22_, the faster the cooling rate of meat.

### 3.6. Scanning Electron Microscopy

SEM imaging can be used to assess the microstructures of meat or meat products [22,34]. The SEM images of the drumsticks under 100× magnification with different cooling treatments are presented in Figure 5. Compared with the IVC drumsticks, the VC drumsticks presented a more disordered muscle fiber structure, as well as an increase in pore size and number. During VC, the fast movement of the moisture changes the internal structure of the cooked meat, causing muscle fiber separation and the formation of large intercellular spaces [35]. However, during IVC, the solution surrounding the drumsticks prevents water evaporation and reduces the force between the muscle fibers.

**Figure 5 foods-11-04063-f005:**
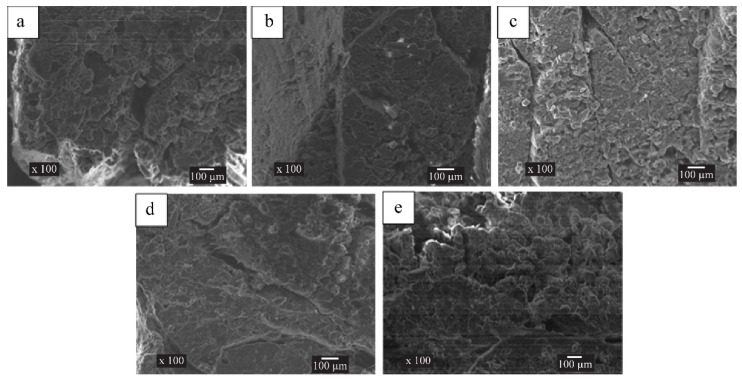
Scanning electron microscopy images showing chicken drumstick muscles that received different cooling treatments. (**a**) Vacuum cooling; (**b**–**e**) immersion vacuum cooling in salt ion concentrations of 0%, 3%, 5%, and 7%, respectively. Scale bars: 100 µm.

The samples cooled by IVC with salt ion concentrations of 0% and 3% in the immersion solution exhibited an ordered muscle fiber structure (Figure 5b,c); the pores mainly exist between muscle fibers and muscle bundles. At the early stage of IVC, the generation of water vapor in drumsticks can promote the growth of pores [35]. When the salt concentration in meat is lower than or close to the salt concentration in brine, the force generated by the pressure difference at the late cooling stage is not enough to destroy the more regular three-dimensional network structure between muscle fibers. However, the ordered structure was disturbed when the salt ion concentration approached 5% (Figure 5d). As the salt concentration increases to 5%, the pores multiply due to the large amount of water vapor at the violent stage of IVC. During the subsequent cooling process, more salt ions penetrate the flesh and muscle tissue, and the proteins and other substances in the drumsticks may be dissolved or denatured due to the presence of salt ions. The uniformity and integrity of the internal muscle tissues were slightly destroyed, and the pores between the muscle fibers become larger. In general, the internal pores of 5% IVC samples became greater in number and the pore size increased. Separation began to occur between some muscle fibers. In addition, the increase in free water content also weakens the cohesiveness between muscle tissues [36], resulting in increasingly larger pores. When the salt concentration was 7%, the number of pores between muscle fibers increased and the pore size increased. The SEM images showed that the structure of some muscle fibers began to become irregular, the boundary of muscle fibers began to blur, the complexity of pores and the tortuous degree of pores increased and the separation of muscle fibers was obvious (Figure 5e). This phenomenon can well explain the decrease in the proportion of free water, which indicates that the flow and transformation of free and bound water in the pores begin to be restricted. The trend of SEM can also explain the variation of cooling rate of IVC samples with different salt ion concentrations.

## 4. Conclusions

This study shows that the regulation of salt ion concentrations in immersion solutions has a significant influence on the cooling time, cooling loss rate, texture, and microstructure of chicken drumsticks. Specifically, higher salt ion concentrations were found to create more irregularities in the drumstick microstructures and pores between the muscle fibers, which enabled the movement of water between the surrounding solution and the inner drumstick, and this significantly altered the cooling time, cooling loss rate, color, and textural properties.

In this study, the brine concentration of drumsticks was 3%, and in IVC, an immersion solution with 5% salt ion concentration could accelerate the cooling rate of drumsticks and increase the meat weight by 2.3%. After cooling, the drumsticks had moderate hardness, a bright yellow color, and good taste. These results indicate that once an optimum salt ion concentration is identified for the immersion solution, the IVC time could be shortened without compromising the drumstick quality. As a follow-up of this research, we will focus on the influence of the vacuum cooling method used in this study on microbial levels in meat.

## Figures and Tables

**Figure 1 foods-11-04063-f001:**
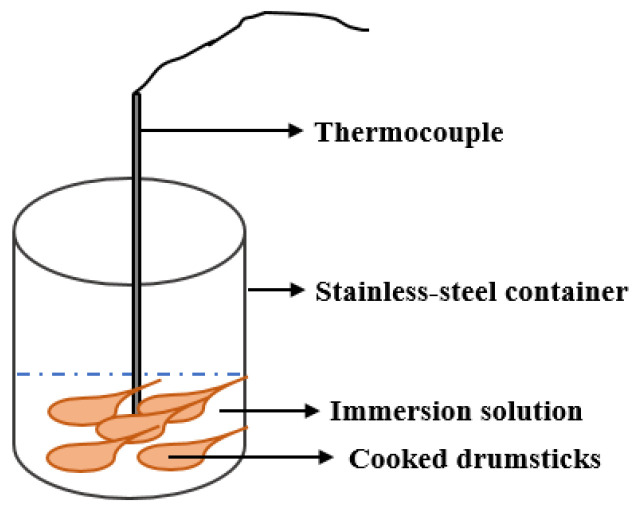
Immersion vacuum cooling equipment.

**Figure 2 foods-11-04063-f002:**
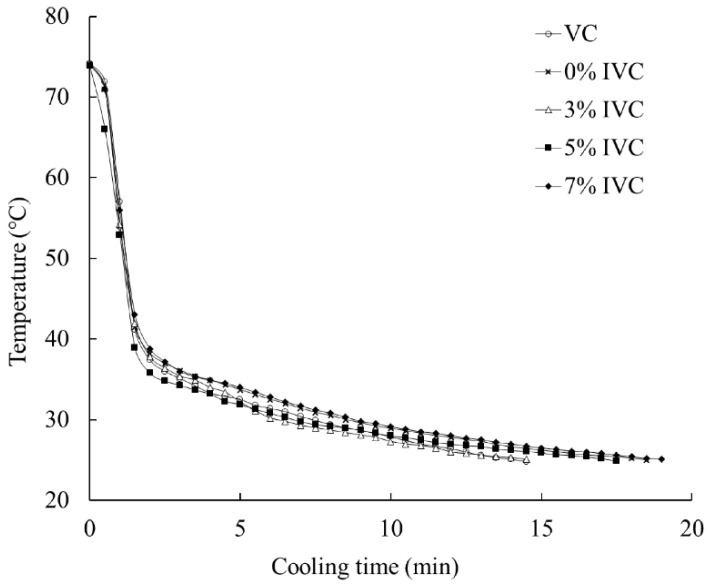
Core temperature curves for chicken drumsticks immersed in solutions with different salt ion concentrations. VC: vacuum cooling; IVC: immersion vacuum cooling; 0%, 3%, 5%, and 7%: salt ion concentrations.

**Figure 3 foods-11-04063-f003:**
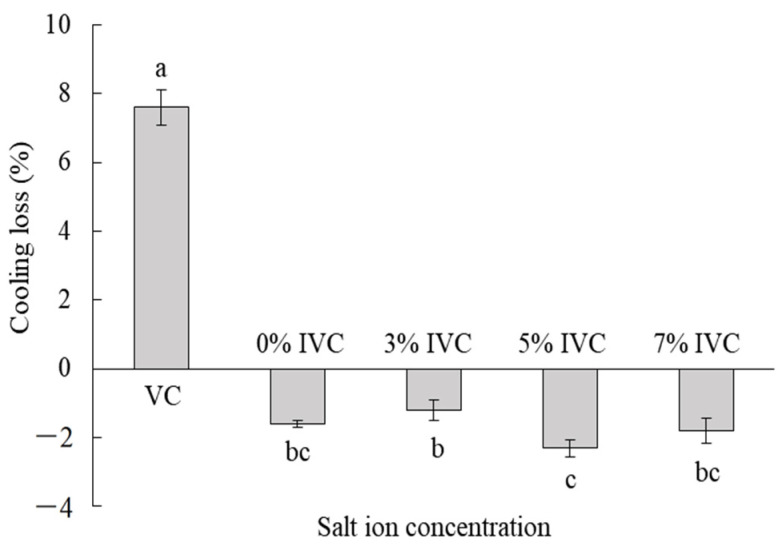
Effects of different immersion salt ion concentrations on the cooling loss of chicken drumsticks. VC: vacuum cooling; IVC: immersion vacuum cooling; 0%, 3%, 5%, and 7%: different salt ion concentrations. The data in the figure are expressed as means ± standard deviation. Different lower-case letters indicate significant differences between different groups. Positive values represent weight loss, whereas negative values represent weight gain.

**Table 1 foods-11-04063-t001:** Parameters for the instruments used to measure the textural properties of the cooled chicken drumsticks.

Probes	P/36R ^1^	A/MORS
Pre-test speed (mm/s)	1.0	2.0
Test speed (mm/s)	4.0	2.0
Post-test speed (mm/s)	4.0	10.0
Compression rate (%)	30	-
Time interval (s)	5.0	-
Cutting depth (mm)	-	5.0

^1^ P/36R and A/MORS are the probe models. “–“ means this indicator does not need to be set.

**Table 2 foods-11-04063-t002:** Effects of immersion solutions with different salt ion concentrations on drumstick color.

Salt Ion Concentrations ^1^	*L**	*a**	*b**	Δ*E*
VC	68.44 ± 1.34 ^a^	4.60 ± 0.47 ^a^	21.12 ± 0.97 ^a^	6.32 ± 0.37 ^a^
0%	73.53 ± 1.52 ^b^	4.63 ± 0.69 ^a^	17.73 ± 0.71 ^b^	2.19 ± 0.19 ^c^
3%	72.98 ± 1.24 ^b^	4.47 ± 0.25 ^a^	18.69 ± 1.60 ^b^	3.28± 0.22 ^b^
5%	73.12 ± 0.55 ^b^	4.41 ± 0.76 ^a^	17.06 ± 0.40 ^b^	2.11± 0.15 ^c^
7%	74.33 ± 1.05 ^b^	4.69 ± 0.87 ^a^	17.61 ± 0.31 ^b^	1.96± 0.13 ^c^

^1^ Salt ion concentrations in the immersion solution during immersion vacuum cooling; VC: vacuum cooling. Data are expressed as the mean ± standard deviation for three replicates (*n* = 3). Different superscript lowercase letters in the same column indicate statistically significant differences (Tukey’s test, *p* < 0.05).

**Table 3 foods-11-04063-t003:** Effects of immersion solutions with different salt ion concentrations on the textural properties of drumsticks.

Salt Ion Concentrations ^1^	Hardness (N)	Elasticity (Ratio)	Chewiness (g·s)
VC	639.31 ± 34.88 ^a^	0.87 ± 0.06 ^a^	857.95 ± 50.24 ^a^
0%	532.50 ± 15.94 ^b^	0.81 ± 0.05 ^a^	726.73 ± 64.21 ^b^
3%	508.69 ± 56.89 ^bc^	0.80 ± 0.03 ^a^	740.05 ± 66.47 ^b^
5%	479.92 ± 32.09 ^c^	0.84 ± 0.05 ^a^	690.99 ± 44.24 ^b^
7%	475.01 ± 28.83 ^c^	0.83 ± 0.02 ^a^	741.79 ± 46.68 ^b^

^1^ Salt ion concentrations in the immersion solutions during immersion vacuum cooling; VC: vacuum cooling. Data are expressed as the mean ± standard deviation of three replicates (*n* = 3). Different superscript lowercase letters in the same column indicate statistically significant differences (Tukey’s test, *p* < 0.05).

## Data Availability

The data presented in this study are available on request from the corresponding authors.

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
