# Peer review of "Effects of Different Salt Ion Concentrations in Immersion Vacuum Cooling on the Qualities of Spiced Chicken Drumsticks"

_foods, 2022, doi:10.3390/foods11244063_

Round 1
Reviewer 1 Report
Dear Authors,
The manuscript is well design and presented therefore, there are few corrections should be need before further processing of your manuscript.
1. Should be highlight the best value of your research in abstract section
2. Check grammatical/typo errors throughout the manuscript
2. Authors needs to avoid keywords, those already mentioned in title of the study.
Reviewer 2 Report
The manuscript entitled "Effects of Different Salt Ion Concentrations in Immersion Vac-2 uum Cooling on the Qualities of Spiced Chicken Drumsticks" focused on effects of different salt ion concentrations in the immersion solution on the cooling rate, cooling loss rate, chromatism, and textural properties of stewed chicken drumstick and determination the mechanisms of water exchange and structural changes.Although the manuscript has written well and professionally and contain some merit points but there are some points needs more attention as below:
Introduction have to improve to illustrate the gap of research in this field, importance and novelty of current paper.
Introduction is containing irrelevant issues and missed important background in this field.
Explain more details about Immersion vacuum cooling and Low-field nuclear magnetic resonance methods.
How did authors perform the quality assurance of data?
Explain more about findings of Table 2 and figure 2.
It is recommended to consider precisely on effects of immersion solutions with different salt ion concentrations on the textural properties of drumsticks results and interpretation of them.
Was there any correlation between color and textural properties in findings?
Discussion of Low-field nuclear magnetic resonance, transverse relaxation time and Scanning electron microscopy analysis needs to improve through interpretation of results and comparison to update references.
It is recommended to rewrite conclusion after discussion revise and improvement.
Consider more on punctuation of literature.
Reviewer 3 Report
The work is very interesting, but requires a few corrections. The abstract could include quantitative results of the research.
Introduction: Please provide examples of application of immersion vacuum cooling and influence of IVC on physicochemical properties of products.
Materials and methods: Where did the research material come from? How many chicken drumstick were used for the experiment.
L95: Why was the temperature reduced only to 25C?
L100, L112: What weight was used for the measurements?
L171: Add literature to the discussion of the results.
The presented results of the work are not sufficiently compared with the literature - restlts of color, texture.
L232: You can count the overall difference in color: deltaE
Conclusions section - add quantitative results.
Reviewer 4 Report
The authors evaluated the effect of a new immersion vacuum cooling on the quality of a ready-to-eat meat product; and compare the new method with the conventional method. The tests are enough and the manuscript is novel. However, there are some points that should be considered and corrected:
-Line 19: The non-significant result (results) is not well specified. Which factor caused the non-significant result?
-Line 54: “cling” may be “cooling”. Please correct it.
-Line 67–69: The sentence should be re-written.
-Line 82: It is not clear that the concentrations of added salt were w/v or w/w. It should be mentioned in the text.
-Lines 88–91: The sentence should be corrected for its grammar and structure. I advise to broke it to 2 or 3 sentences.
- It is not necessary to repeat the number of replicates in each section. Please omit the repeated sentences in various sections (for example when the authors describe the test methods 2.7, 2.8, 2.9, …).
-Section 2.8: Did the authors considered blooming time when assessed the instrumental color values?
-Section 2.12: Which type of ANOVA was used for statistical analysis. One-way, two-way, or repeated measure? It should be specified in the manuscript.
-Line 184–198 and 220– 231: Please discuss about your results. Compare your results with the results of the other researches in this field.
-Line 248–249: Please add a reference (or references) for this sentence.
-Line 250: The sentence should be corrected for its grammar.
-Conclusion section: Please re-write the last part of this section by giving the core message to food industry.
Round 2
Reviewer 2 Report
The revised of manuscript entitled" Effects of Different Salt Ion Concentrations in Immersion Vacuum Cooling on the Qualities of Spiced Chicken Drumsticks" has improved significantly.
Though most of necessary changes have performed, some points in discussion section have to consider more as below:
1- Low-field Nuclear Magnetic Resonance results have to interpret properly.
2- The transverse relaxation time (T2) of drumsticks results interpretation needs improvement.
3- Results related to images of scanning electron microscopy needs discus more.
Reviewer 3 Report
The authors improved their manuscript.
Author Response
Thanks a lot for your suggestion and professional guidance on this article.